

# Empirical high-resolution wind field and gust model in mountainous and hilly terrain based on the dense WegenerNet station networks

Christoph Schlager[1], Gottfried Kirchengast[1], and Juergen Fuchsberger[1]

[1]Wegener Center for Climate and Global Change (WEGC), and Institute for Geophysics, Astrophysics, and Meteorology/Institute of Physics, University of Graz, Graz, Austria.

*Correspondence to:* Christoph Schlager (christoph.schlager@uni-graz.at)

**Abstract.** A weather diagnostic application for automatic generation of gridded wind fields in near-real time, recently developed by the authors (Schlager et al., 2017), is applied to the WegenerNet Johnsbachtal (JBT) meteorological station network. This station network contains eleven meteorological stations at elevations from about 600 m to 2200 m in a mountainous region in the north of Styria, Austria. The application generates, based on meteorological observations with a temporal reso-

lution of 10 minutes from the WegenerNet JBT, mean wind and wind gust fields at 10 m and 50 m height levels with a high spatial resolution of 100 x 100 m and a temporal resolution of 30 minutes. These wind field products are automatically stored to the WegenerNet data archives, which also include long-term averaged weather and climate datasets from post-processing. A main purpose of these empirically modeled products is the evaluation of convection-permitting dynamical climate models as well as investigating weather and climate variability on a local scale. The application's performance is evaluated against

the observations from meteorological stations for representative weather conditions, for a month including mainly thermally induced wind events (July 2014) and a month with frequently occurring strong wind events (December 2013). The overall statistical agreement, estimated for the vector-mean wind speed, shows a reasonably good modeling performance with somewhat better values for the strong wind conditions. The difference between modeled and observed wind directions depends on the station location, where locations along mountain slopes are particularly challenging. Furthermore, the seasonal statistical

agreement was investigated from five-year climate data of the WegenerNet JBT in comparison to nine-year climate data from the high-density WegenerNet meteorological station network Feldbach Region (FBR) analyzed by Schlager et al. (2017). In general, the five-year statistical evaluation for the JBT indicates similar performance as the shorter-term evaluations of the two representative months. Because of the denser WegenerNet FBR network, the statistical results show better performance for this station network. The application can now serve as a valuable tool for intercomparison with and evaluation of wind fields from

high-resolution dynamical climate models in both the WegenerNet FBR and JBT regions.



# 1  Introduction

Advances in computer sciences and the growing power of computers enable meanwhile highly-resolved model outputs from regional climate models (RCMs) with horizontal resolutions at a scale of 1 km. At this resolution RCMs provide more realistic simulations, especially for regions with complex terrain, and allow to investigate weather and climate in small sub regions (Awan et al., 2011; Suklitsch et al., 2011; Prein et al., 2013, 2015; Leutwyler et al., 2016; Kendon et al., 2017).

To evaluate RCMs and to improve the performance of such models, meteorological observations and particularly gridded datasets in correspondingly high spatial and temporal resolutions are needed. RCMs generally represent processes area-averaged rather than on a point-scale (Osborn and Hulme, 1998; Prein et al., 2015). Therefore, gridded fields of meteorological data are the most appropriate evaluation datasets, with each grid value being a best estimate average of the grid cell observations (Haylock et al., 2008; Haiden et al., 2011; Hiebl and Frei, 2016).

For investigating weather and climate on a local scale as well as evaluating RCMs, the Wegener Center (WEGC) at the University of Graz acquires and automatically processes data from two station networks: the WegenerNet Feldbach Region (FBR) and the WegenerNet Johnsbachtal (JBT). The WegenerNet FBR is located in south-eastern Styria, Austria and covers a dense grid of more than 150 meteorological stations within an area of about 22 km x 16 km (Kirchengast et al., 2014). The terrain of the FBR is hilly and characterized by small differences in altitude, and the region is quite sensitive to climate change (Kabas et al., 2011; Kabas, 2012; Hohmann et al., 2017). It exhibits rich weather variability, especially including strong convective activity and severe weather in summer (Kirchengast et al., 2014; Kann et al., 2015; O et al., 2017, 2018). Recently Schlager et al. (2017) also analyzed wind fields in this region.

The focus of this study is on the WegenerNet JBT, a station network consisting of eleven meteorological stations, located in a mountainous region in upper Styria, which is characterized through a very complex terrain (Fig. 1). The WegenerNet JBT has been realized through an interdisciplinary research cooperation and the stations are operated by the WEGC and several different partner organizations (indicated in Fig. 1). Details related to the cooperation, partnerships and first results can be found in Strasser et al. (2013).

All observations from the two WegenerNet regions are integrated into the WegenerNet Processing System (WPS), a system to control and manage meteorological station data (Kirchengast et al., 2014). This WPS consists of four subsystems: The Command Receive Archiving System transfers raw data via General Packet Radio Service (GPRS) wireless transmission to the WegenerNet database in Graz, the Quality Control System checks the data quality, the Data Product Generator (DPG) produces regular station time series and gridded fields of weather and climate products, and the Visualization and Information System offers the data to users via the WegenerNet data portal (www.wegenernet.org).

The DPG-produced weather and climate products are stored to the WegenerNet data archives and include since many years already the gridded fields of the variables temperature, precipitation and relative humidity for the WegenerNet FBR. These fields are generated based on a spatial interpolation of the station observations and provided with a latency of about 1-2 hours. Temperature lapse rates estimated from the observational datasets at the many different station altitudes are included in the generation of temperature fields over the hilly terrain. Technically, for temperature and relative humidity, the fields



are constructed by an inverse-distance weighted interpolation and for the precipitation the inverse-distance squared weighted interpolation is used. Details related to the subsystems of the WPS can be found in Kirchengast et al. (2014) and Kabas (2012).

Furthermore, since the recent work of Schlager et al. (2017), the DPG computes spatially distributed wind fields for the WegenerNet FBR. Due to the dependence of wind on many different conditions, including surface properties such as topography

and surface roughness, we use a newly developed application (named Wind Product Generator or WPG, developed in Python) to determine the gridded field of wind parameters (Abdel-Aal et al., 2009; Sfetsos, 2002; Schlager et al., 2017).

The WPG uses the freely available empirical California Meteorological Model (CALMET) as core tool and generates wind fields in near real-time. The CALMET model reconstructs 3D wind fields (we focus on the 10 m and 50 m height levels) based on meteorological observations, terrain elevations and information about land usage. Before its routine use for the

WegenerNet FBR, the WPG has been evaluated for a month including mainly thermally induced events and another month with frequently occurring strong wind events; the statistics showed good results for these periods. A detailed description of the WPG application, and the statistical results for the WegenerNet FBR, can be found in Schlager et al. (2017).

The key goal of this study is the implementation and evaluation of the WPG to automatically produce high-resolution wind fields in near real time also for the second study area, the challenging WegenerNet JBT region with its terrain from about 700 m

to 2300 m and less wind stations than for the WegenerNet FBR. The requirement for our WPG application is to provide the JBT wind fields also with a spatial resolution of 100 m x 100 m and a time resolution of 30 min to the WegenerNet data archives. An essential goal is the evaluation of these wind fields for periods with representative weather conditions and also the estimation of wind gust fields. Furthermore, the WPG's performance shall be estimated first-time also for seasonal climate-averaged data for the WegenerNet JBT in comparison to the WegenerNet FBR region.

The paper is structured as follows. Section 2 provides a description of the study area, the WegenerNet JBT region with its meteorological stations. Section 3 presents the methodology for the empirical wind field modeling, where first the characteristics of the CALMET model and the extensions integrated to the WPG (Schlager et al., 2017) for the automated production of the wind fields are explained, in particular the inclusion of a few auxiliary pseudo stations (Fig. 1). Second, the estimation method for the gust fields and a description of atmospheric weather conditions during the model evaluation periods and of the

evaluation methods is introduced here. Section 4 describes the results of the wind field modeling for the selected evaluation periods, July 2014 and December 2013, for the WegenerNet JBT as well as the results of the seasonal climate datasets from the WegenerNet JBT compared to those of the WegenerNet FBR. Finally, section 5 provides the conclusions as well as prospects for the next steps of follow-on work.

## 2 Study Area and WegenerNet Data

The study area WegenerNet JBT (Fig. 1), named after the Johnsbach river basin, is located in the *Ennstaler Alps*, an eastern Alpine region in the north of Styria, Austria, and overlays with the *National Park Gesäuse*. The area is surrounded by the *Gesäuse Mountains* in the north, east and west and by the *Eisenerzer Alps* in the south. The terrain is characterized by large differences in elevation, ranging from below 700 m in the valleys to over 2.300 m at the highest summits (Strasser et al., 2013).



The highest peak is the Hochtor, with an elevation of 2369 m. The landscape is dominated by alpine rock formations and sparsely vegetated areas (barren land), forests, and range land (Fig. 2a).

The climate is Alpine with annual mean temperatures of around 8 °C at lower elevations and below 0 °C at higher elevations and with an annual precipitation of about 1.500 mm to 1.800 mm from the valley to the summit regions (Wakonigg, 1978;

Prettenthaler et al., 2010). The summer-day temperature field illustrated in Fig. 2b, produced by a modified version of CALMET (Schlager et al., 2017), shows the distinct decrease in temperatures from lower to higher elevations. We implemented algorithms developed by Bellasio et al. (2005) as part of this modified CALMET version to account for topographic shading and height dependency in surface temperatures (more details in Section 3). The wind field in the study area is characterized by thermally induced local flows and influenced from larger scales mainly by westerly-flow synoptic weather conditions.

The WegenerNet JBT comprises eleven irregularly distributed meteorological stations within its area of about 16 km x 17 km. The station with the highest altitude was installed in summer 2009 and is located on the summit of the Zinoedl at a height of 2.191 m. A second summit station was installed in 2011 on top of the Blaseneck at a height of 1.969 m (Strasser et al., 2013).

All stations are equipped with a diversity of meteorological sensors. The observed variables wind speed ($v$), wind direction ($\phi$), air temperature ($T$), air pressure ($p$) and relative humidity ($rh$) are continuously recorded at a 10 min sampling rate and

used as input for the WPG. Table 1 summarizes the technical characteristics of the WegenerNet JBT stations including the station operators, wind sensor heights, and observed variables for each station (including the ones used). Due to a topography strongly influencing the local wind fields at the Weidendom and the Tamischbachturm 1 stations, the observations of the wind variables from these two stations are not used as input.

The observations of the Wegener JBT stations are partly available since 2010, and partly since 2007 (Table 1, first column).

For this study, wind fields have been calculated within the period 2012 – 2017, and ongoing near-real-time data are to be provided to the users with a maximum delay of 2 hours.

## 3 Methods and evaluation periods

### 3.1 Advanced CALMET model

The core tool of the operational WPG is the CALMET model (Scire et al., 1998). Based on the settings in the CALMET

control file, a user has three different options for the use of the meteorological information as input data: in the no-observations approach, CALMET uses data from numerical prognostic models as input data, the hybrid approach combines data from numerical models and meteorological observations, and the observations-only approach solely uses meteorological observations. We use the observations-only approach for the WPG, to ensure genuinely empirical wind fields and to keep the key operational input independent from data external to the WegenerNet (Schlager et al., 2017; Scire et al., 1998). We consider this also the

best-possible choice for later intercomparison to and evaluation of dynamical climate model fields.

The CALMET model computes the wind fields in a two-step approach. The first step (step 1) includes the adjustment of an initial-guess wind field for kinematic effects of terrain, slope flows, and terrain blocking effects. In the observations-only approach the initial-guess wind field is produced by an interpolation of observational data.





In a second step (step 2), the observational data are introduced again and blended to the step 1 wind field by an inverse distance weighting interpolation to produce the final step 2 wind field. Observations are excluded from this interpolation method if the distance from a station location to a particular grid point is greater than a user defined radius of influence. Furthermore, relative weighting parameters are used to weight the observed wind variables to the previously computed step 1 wind field (Table 2). The procedure ensures divergence-free (mass-conserving) wind vector fields, i.e., provides physically consitent fields under assumption of incompressible flow.

Based on extensive sensitivity tests, we determined the settings for the WegenerNet JBT shown in Table 2. Comparing these to the settings of Schlager et al. (2017), Table 2 therein, for the WegenerNet FBR, one can see that we in particular found it beneficial to increase the influence of terrain features and the first-guess file in the surface layer. A detailed description related to model parameters, settings and options can be found in the CALMET Manual (Scire et al., 1998).

In the original CALMET model, the energy balance is calculated without considering topographic shading through terrain. Furthermore, height dependency of surface temperatures is not taken into account and the temperature fields are produced by a simple interpolation of point-specific observations. Especially in complex terrain like in the WegenerNet JBT, such shading, vertical temperature gradients and the vegetation cover significantly affect the energy balance and subsequently the wind field.

To improve the modeling of these physical effects in this challenging region, we improved an advanced model by implementing algorithms developed by Bellasio et al. (2005). These algorithms empirically take into account the topographic shading based on terrain heights, topography slope and aspect, and the position of the sun for the estimation of solar radiation. In addition, temperature fields are modeled based on vertical temperature gradients, estimated from the meteorological stations located at different altitudes, and the influence of the vegetation cover is accounted for, based on the leaf area index (LAI) obtained from a geophysical dataset (Table 3). Detailed information related to these algorithms can be found in Bellasio et al. (2005).

The WPG runs this advanced CALMET model based on a surface meteorological data file, upper air data files and a geophysical data file. In a predecessor step, the WPG automatically generates these meteorological data sets from the station observations and auxiliary geophysical information stored in the WegenerNet database. Detailed information related to the WPG, including all processing steps, can be found in Schlager et al. (2017).

The geophysical dataset consists of terrain elevations and land use categories and was created in a preparatory step. In this study we used a DEM derived from airborne laser scanning point clouds (provided online by http://gis.steiermark.at), illustrated by the elevations scale in Fig. 1a. The original spatial resolution of 10 m was resampled and averaged to 50 m (DEM50), 100 m (DEM100), and 200 m (DEM200). In order to check the influence of the spatial resolution on the modeling performance, the model was tested with the different spatial resolutions. These sensitivity tests showed very small differences between wind field results modeled based on DEM50 and DEM100, while somewhat higher differences (from smoothing effects) were found when using DEM200. We hence selected as the most adequate and computationally efficient resolution the DEM100 and the 100 m x 100 m gridding for this study, which also matches the resolution of the land cover dataset discussed next.

Furthermore, the land use categories for the study were determined based on the Corine Land Cover 2006 dataset (CLC 2006) (EEA, 2007). The definition and the maximum number of the land use categories of the CLC dataset differs from the





classification scheme of the CALMET model. The entire CLC dataset of the third and most detailed level contains 44 different classes, while the CALMET classification scheme only distinguishes up to 14 land use types (Oleniacz and Rzeszutek, 2014). We therefore reclassified the 17 CLC 2006 land use categories found in the study area into seven compliant CALMET classes (Fig. 2a); the corresponding parameters summarized in Table 3 were then used as the CALMET geophysical dataset.

The observations of the three highest stations Zinoedl, Blaseneck and Tamischbachturm 2 (Table 1) are used to create vertical profiles of wind speed, wind direction, temperature, pressure, and elevation, stored in upper air datasets. A detailed explanation of how the creation of upper air datasets works can be found in Schlager et al. (2017).

## 3.2 Auxiliary pseudo stations for upper-air data

Based on finding a systematic underestimation of wind speed in summit regions without any station, we extended the WPG
with a user option that enables to introduce pseudo upper-air stations in the modeling domain. These user-defined pseudo stations are included to raise wind speed at higher altitudes. For the WegenerNet JBT we defined five pseudo stations upon extensive sensitivity studies testing various setups, located at the unobserved summit regions (Table 4, Fig. 1 and Fig. 2). The magnitude of wind speed of a pseudo station ($v_p$) is estimated for the highest defined vertical height level ($z_{max}$), which corresponds to the highest $ZFACE$ level (Table 2), by linear interpolation between neighbor station altitudes (resp. a slight
downward extrapolation for pseudo station 5)

$$v_p(z_{max}) = v_{n1}(z_{max}) + \left[ \frac{v_{n2}(z_{max}) - v_{n1}(z_{max})}{z_{n2} - z_{n1}} \right] (z_p - z_{n1}), \tag{1}$$

where $z_p$ is the altitude of the pseudo station and $z_{n1}$ and $z_{n2}$ indicates the altitudes of the defined neighbor stations with real wind observations (Table 4, rightmost column).

    The magnitude of the wind speeds $v_{n1,2}(z_{max})$ at the highest height level of the neighbor stations used in Eq. (1) are
calculated by a logarithmic wind profile given as

$$v_{n1,2}(z_{max}) = v_{n1,2}(z_{s1,2}) \frac{\ln(z_{max}/z_0)}{\ln(z_{s1,2}/z_0)}, \tag{2}$$

where $v_{n1,2}(z_{s1,2})$ are the wind speeds at the neighbor stations observed at the sensor heights $z_{s1,2}$ (typically 5-10 m above surface), and $z_0$ is the surface roughness length at the locations of the corresponding neighbor stations (up to 1 m).



The wind direction at the pseudo station $\phi_p(z_{max})$ is estimated through a vector-mean calculation by

$$\phi_p(z_{max}) = \begin{cases} 0° & \text{when } v \geq 0 \ \& \ u = 0 \\ 180° & \text{when } v < 0 \ \& \ u = 0 \\ 90° - \arctan \sqrt{(\frac{v}{u})^2} & \text{when } v \geq 0 \ \& \ u > 0 \\ 90° + \arctan \sqrt{(\frac{v}{u})^2} & \text{when } v \leq 0 \ \& \ u > 0 \\ 270° - \arctan \sqrt{(\frac{v}{u})^2} & \text{when } v \leq 0 \ \& \ u < 0 \\ 270° + \arctan \sqrt{(\frac{v}{u})^2} & \text{when } v \geq 0 \ \& \ u < 0 \end{cases}, \tag{3}$$

where the mean values of the north component $v$ and the east component $u$ are calculated from the wind directions at the two neighbor stations by

$$v(z_{max}) = \frac{1}{2}[\cos\phi_{n,1}(z_{max}) + \cos\phi_{n,2}(z_{max})], \tag{4}$$

and

$$u(z_{max}) = \frac{1}{2}[\sin\phi_{n,1}(z_{max}) + \sin\phi_{n,2}(z_{max})]). \tag{5}$$

For providing $\phi_{n1,2}$ to these equations, the wind directions at the neighbor stations are extrapolated to $z_{max}$ based on the work of van Ulden and Holtslag (1985) by

$$\phi_{n1,2}(z_{max}) = \phi_{n1,2}(z_{s1,2}) \, \Theta \, d_1 \left[1 - \exp\left(-d_2 \frac{z_{max}}{z_{s1,2}}\right)\right], \tag{6}$$

where $\phi_{n1,2}(z_{s1,2})$ are the observed wind directions at the neighbor stations at the sensor heights $z_{s1,2}$. The empirical constants $d_1$ and $d_2$ take the values 1.5 and 1.0, respectively. For this extrapolation we assume neutral stability conditions, which means the turning angle $\Theta$ is set to 12°. Details can be found in van Ulden and Holtslag (1985) and the CALMET user guide (Scire et al., 1998).

Eq. (2) and Eq. (6) are then used again, but in this case to compute the wind speed and wind direction at the pseudo stations (Table 4) for the defined height levels, based on the values estimated at $z_{max}$ from Eq. (1) and Eq. (3).

The temperatures at the pseudo stations are estimated from the gridded temperature field generated by an interpolation of the temperature observations. To calculate the temperatures for the defined station altitudes and height levels, temperature lapse rates are estimated from the temperature observations of the meteorological stations; for the relevant details on the generation of the upper-air datasets see Schlager et al. (2017).

An additional user option that we integrated into the WPG concerns the replacement of missing observations from meteorological stations that are used to create the upper-air datasets. If observations from such a station show invalid values, indicated





by quality flags, the WPG includes an algorithm to replace these data with observations from the highest upper air station with valid wind data. To indicate the data quality to the users, we additionally provide gridded quality flags, ranging from zero (good value) to four (bad value).

### 3.3 Wind gust fields as add-on product

As an additional post-processed product, we let the WPG generate gridded fields of peak gust speed and the corresponding gust direction for 10 m height above ground, based on re-scaling the gridded mean wind fields with the aid of complementary wind gust observations ($v_g$, $\phi_g$) of the meteorological stations (Table 1). While a detailed evaluation of this add-on product is beyond the scope of this study it fits to briefly introduce its generation and some example results here, because these gust fields are since recently also routinely available via the WegenerNet data portal www.wegenernet.org.

More specifically, the gridded gust speeds are generated by a spatial interpolation of the ratio of the observed maximum 30 min gust speed to the 30 min average wind speed, where this speed ratio is determined at each observing station location by

$$r_{gm}^v = \frac{v_g}{v_m}, \tag{7}$$

where $v_g$ is the peak gust speed and $v_m$ the average wind speed. The ratio field, generated by interpolating $r_{gm}^v$, is then multiplied to the gridded mean speed field, yielding a gridded gust field. As interpolation method for the wind speed ratio, a

simple inverse distance algorithm is employed in the WPG, which leads to a reasonably smooth gridded gust-to-mean ratio field. This procedure is a rough but sound approximation of how strong in any 30 min time slice the wind gustiness is pronounced, on top of the prevailing mean wind speeds.

    To generate the gridded wind gust directions, the approach is essentially the same but with using direction differences instead of speed ratios. That is, the WPG determines the difference between the gust direction of the peak gust speed and the 30 min

vector-mean wind direction. This wind difference is computed by

$$\Delta\Phi_{gm} = \Phi_g - \Phi_m, \tag{8}$$

where $\Phi_g$ is the direction of the peak gust speed and $\Phi_g$ the 30-min vector-mean wind direction. The spatial interpolation of these direction differences ($\Delta\Phi_{gm}$) to the grid is done as for the gridded speed ratios. As interpolation method again a a simple inverse distance algorithm is employed. Finally, these gridded direction difference fields are added to the mean wind direction

fields to obtain the wind gust direction field.

### 3.4 Wind field evaluation periods

The modeling performance is first evaluated by periods with mainly two representative types of wind events: thermally induced wind events and strong wind events. We have chosen July 2014 and December 2013 as test months for this purpose.



In July 2014 the study area was mainly controlled by autochthonous weather conditions, characterized by small synoptic influences, cloudless sky, low relative humidity and increased radiation fluxes between the Earth surface and the atmosphere (Prettenthaler et al., 2010). These weather conditions mainly led to thermally induced wind systems, meaning that the wind fields were dominated by small-scale temperature and pressure gradients. In December 2013 several episodes of strong wind

occurred, including wind storms with 30-min wind speeds up to around 30 m s$^{-1}$ and peak gusts up to 55 m s$^{-1}$ . Wind speeds $< 0.5\ m\ s^{-1}$ were classified *calm* and discarded as to small for a reliable cross-validation.

For estimating the model performance we used a leave-one-out cross-validation, as in our previous Schlager et al. (2017) work. In this methodology, wind observations at one wind station are removed from the stations input to the WPG and generated wind fields are evaluated against the wind data from this station. More specifically, we compared the output wind field

results at the station location with the observations of the respective station for each 30-min sample. We then calculated the statistical performance parameters summarized in Table 5 from all data over the full evaluation period, for all seven stations that contributed wind sensors (all wind observing stations in Table 1 except WEI and TA1).

Regarding the index of agreement (*IOA*) parameter we note that in this study we used a redefined *IOA*, which spans from -1 to +1 with values near +1 indicating best model performance (Willmott et al., 2012). An IOA of 0.5, for example, implies that

the sum of the difference magnitudes between modeled and observed values is one-half of the sum of the observed deviation magnitudes. An opposite value of -0.5 indicates that the sum of the difference magnitudes is twice the sum of the observed deviation magnitudes. In case of little observed variability or poorly estimated deviations about $\overline{v_o}$, the *IOA* delivers a value near -1.

In addition, we calculated statistical performance parameters for five-year seasonal data of the WegenerNet JBT and com-

pared the results to nine-year seasonal data of the WegenerNet FBR. We used the WegenerNet independent wind measurements from the ZAMG Feldbach and Bad Gleichenberg stations, located in the FBR, and from the ZAMG Admont station, located near the JBT area (a few kilometers west of it, see Fig. 1) for this climatological evaluation. For the WegenerNet JBT we used, in addition to the ZAMG Admont station, the wind measurements from the representative "left-out" stations KOE and BLA.

## 4   Results

### 4.1   Evaluation of representative summer and winter month

Figure 3 illustrates typical examples of WPG-modeled wind fields for morning (upper panels), afternoon (middle panels), and evening (lower panels) winds at a height of 10 m. The left column (Fig. 3a) shows thermally driven circulations in course of the 18th of July 2014 with varying wind speeds and directions caused by temperature and pressure gradients on a local scale. The highest wind speeds typically occurred in the summit regions, with maximum 30-min wind speeds of around 7 m s$^{-1}$ near

sunrise at 04:00 UTC (05:00 LT).

The right column (Fig. 3b) displays wind storm behavior during the 7th of December 2013 caused by northwesterly weather conditions. These synoptic-scale flow conditions led to strong wind speeds in the area with prevailing northwesterly wind directions and maximum 30-min wind speeds of around 30 m s$^{-1}$ during the early morning at 04:00 UTC (Fig. 3b, top).





Later during the day slightly weaker wind speeds occur and the air flow is more influenced by the terrain and partly channeled through the valleys of the study area.

The maps in Fig. 4, shown in the same layout as Fig. 3, display the estimated distribution of the peak gust speeds and the corresponding gust directions for the same days. Note that these are neither instantaneous nor average gust fields but synthetic

field estimates of maximum wind peaks and associated directions that occurred at the some time during the 30-min sample interval. The thermally driven gust field on the 18th of July 2014 showed maximum gust speeds of around 18 m s$^{-1}$ upstream to the Zinoedl summit and the ridge of TA1 at 14:30 UTC (15:30 LT) (Fig. 4a, middle). During the storm event on the 7th of December 2013, the gusts reached a tremendous speed of near 55 m s$^{-1}$ at 04:00 UTC (Fig. 4b, top) around the Zinoedl summit and the summit pseudo station PS2 (around 200 km h$^{-1}$). It is noticeable that the strongest gusts have a northerly

direction whereas the average wind comes from the northwest (Figs. 4b and 3b).

Figures 5 and 6 illustrate the modeling performance at the Koelblwiese (KOE) and the Blaseneck (BLA) station, as typical examples for a valley and a summit station. The KOE station is located in the Johnsbach valley at a height of 860 m to monitor the climate at the valley floor. Especially in fall and winter the environment of this station is often influenced by lakes of cold air. The BLA station is located at a height of 1969 m on the summit of the Blaseneck. The environment of the latter station is

characterized through an exposed high Alpine location were strong wind speeds can occur in all seasons. In the scatter plots we compared the observed 30-min vector-mean wind speeds to the corresponding modeled values of the nearest neighbor gridpoints (located at < 50 m distance).

For the KOE station we estimated a reasonably good model performance with a correlation coefficient $R$ of 0.71 in July 2014 and 0.75 in December 2013. In July 2014 the maximum observed and modeled wind speeds are around 5 m s$^{-1}$ with a slightly

positive bias $B$ between observed and modeled wind speeds (Fig. 5a). In December 2013 the maximum observed wind speeds are around 13 m s$^{-1}$ and the estimated $B$ is slightly negative (Fig. 5b).

The scatter plot for the BLA station indicates a wider spread of the observed and modeled wind speeds compared to the Koelblwiese station (Fig. 6). Regarding the $R$ value we estimated similar good results with a value of 0.69 for July 2014 and 0.71 for December 2013. The mean absolute error of wind direction $MAE_{dir}$ exhibits similar results for both stations and

periods, with values near 40° (except for 59° at KOE in Dec. 2013).

Figures 7 and 8 show windroses of the relative frequency of wind directions divided by wind speed categories from the model compared to the observed wind directions for the KOE and BLA station, respectively.

Regarding the KOE station in July 2014 (Fig. 7a), a shift from the WSW to the WNW sectors can be seen in the modeled results. The observations show about 18 % in the WSW sector, while the model estimates just a few percent in this sector.

Vica versa, the frequency of observed wind directions is 7 % for the WNW sector, while the model shows 23 % in this sector. This shift by about 40° in wind directions is explained by the influence of the Oberkainz (OBK) station which is located in the WNW in a distance of only about 1 km from the KOE station. The magnitude of the wind speed is calculated quite well by the model, with values below 5 m s$^{-1}$ in accordance to the observations.

In December 2013 (Fig. 7b) the main observed wind directions at the KOE station are from the NNE to the E sectors,

however wind directions with high wind speeds can be observed in the westward sectors as well. For this period, the model





estimates a significantly narrower wind directions corridor, with the highest proportion of wind directions in the NW and the ESE sector (each about 22 %). Evidently, the upslope flow conditions (NE sector) cannot be captured well by the available observational information.

Figure 8 illustrates the BLA station results. In July 2014 (Fig. 8a), the observed prevailing wind directions are from the NNW to the ENE sectors, while the model calculates the highest proportion from the WNW to N sector. Regarding wind speed, the model estimates values in good agreement with the observed wind speeds, illustrated in Figure 8 a.

In December 2013 (Fig. 8b) a shift between observed and modeled wind directions from the NNW to the WNW sector and from the SW more to the W sector can be seen. These modeled westerly flows are caused by the influence of the summit station Zinoedl (ZIN), which is mainly driven by northwesterly flows in this period. As briefly explained in Section 3 above, the WPG implements a function to replace missing upper-air data with valid observations from the highest upper air station, giving the reason for the influence of this station. In case of the evaluation of the BLA station the missing upper-air data were replaced by observations from the ZIN station. For this period, again the wind speeds between the observations and the model results are in good overall agreement.

The statistical results from all meteorological stations are summarized in Table 6. The absolute statistical parameters (bias $B$, standard deviation $SD_o$, root-mean-square-error $RMSE$, and mean absolute error of wind direction $MAE_{dir}$) applied to the vector-mean of wind speed show considerably higher values in December 2013, resulting from the overall higher wind speeds in this period. In general, the $B$ values are positive, except for the ZIN station and for the OBK station in July 2014.

The mean $R$ values show better results in December 2013 than in July 2014 and the estimated $MAE_{dir}$ is similar for both periods, and found at near or below about 40°. The $RMSE$ values range from 0.8 to 3 m s$^{-1}$ for July 2014, with the lowest value for the KOE station and the highest value for the TA2 station. The December 2013 generally shows higher $RMSE$ values, with the lowest value (1.35 m s) again for the KOE station and the highest value (6 m s) for the ZIN station.

The $SD_o$ values are of similar size for both periods. The mean $R$ value is 0.58 for July 2014 and 0.69 for December 2013. For December 2013, the $R$ value is higher than 0.6 for all stations except for OBK, compared to July 2014, where all stations show higher values than 0.5, except for OBK and SCH. Regarding the mean $IOA$, we estimated a value of 0.51 for July 2014 and 0.43 for December 2013, with again remarkably low values for the SCH station in July 2014 and for the OBK and SCH station in December 2013.

These overall statistical results, but also the example results shown in Fig. 3 to Fig. 10, well illustrate the useful level of skill but also the evident performance limits that the developed WPG application can provide for empirical wind field modeling based on a small set of seven stations in such a complex terrain as the WegenerNet JBT area.

## 4.2 Evaluation based on multi-year climatological data

Modeled average wind fields for the WegenerNet JBT are presented in the multi-year climatological data of Fig. 9 (top panels), showing five year climate data for the summer and winter season. In summer, the seasonal average wind speed reaches maximum values of around 6 m s$^{-1}$ at the highest summits and generally lower values in the valley regions, with around 3 m s$^{-1}$. The environment of the OBK, KOE and SCH stations exhibits the lowest average wind speeds, directly linked to the observa-





tions of these stations which are used as model input (Fig. 9a, color shading). In comparison, the winter months show generally higher average wind speeds, with a similar spatial distribution but including in particular higher values at higher altitudes and the summit regions. The maximum average wind speeds of around 8 m s$^{-1}$ is observed at the highest summits (Fig. 9b, color shading).

The vector-mean of wind directions for the summer season has directions mainly from the S sectors with maximum vector-mean wind speeds of around 3 m s$^{-1}$ (Fig. 9a, black arrows). In the winter season, the prevailing wind directions are from the W sectors, with maximum vector-mean wind speeds of around 5 m s$^{-1}$ (Fig. 9b, black arrows).

The windroses of Fig. 9 bottoms show the seasonal relative frequency of wind directions for the summer and the winter seasons for the KOE and BLA stations, used as example for a valley and a summit station. The distribution of wind directions

shows similar results as the distribution for the individual month July 2014 and December 2013 (cf. Fig. 7 and Fig. 8). This similar pattern indicates a good representativeness of these months, including evidently common weather conditions in the WegenerNet JBT.

Because of the valley location of the KOE station, the observations and modeled values show narrow wind corridors with a flow mainly along the valley axis during the summer. The largest part of the observed flow is from the directions E to ESE

and WSW to W, while the model estimates directions mainly from the ESE to WNW sector (bottom-left panel of Fig. 9a). In winter, most of the flow is from the NE to the ESE sector. The model again estimates wind directions mainly from the ESE and the WNW to the NW sectors (bottom-left panel of Fig. 9b). A shift between modeled and observed values from WSW to WNW directions can be seen in both seasons; this shift is caused by the observational influence of the nearby OBK station on the modeled wind fields, which is located around 1 km northwest of the KOE station (cf. also Fig. 7).

The relative frequency of observed wind directions of the BLA station shows prevailing directions from the NW to the N in the summer and winter months, while the model mainly estimates wind directions from the W to the NW sectors. In both seasons, the largest fraction is estimated from the WNW sector, with around 12 % in the summer months and around 23 % in the winter months (bottom-right panels of the second row in Fig. 9a and b). The modeled more westerly flows are caused by the influence of the ZIN station; as already indicated by the individual month results of Fig. 8.

For the WegenerNet FBR we show nine-year average wind fields again for the summer and winter season (Fig. 10), in the same format as Fig. 10 shows for WegenerNet JBT. The maximum average wind speeds occur around the highest WegenerNet FBR station 74, located at an elevation of 394 m, with average wind speeds around 1.5 m s$^{-1}$ in summer (Fig. 9a, top) and near 2.0 m s$^{-1}$ in winter (Fig. 10b, top). The spatial distribution of the wind speeds exhibits slightly lower values in summer than in winter. As expected, overall both the modeled average-speed fields and the vector-mean fields from the WegenerNet

FBR (Fig. 10) in the Alpine forelands show much lower wind speeds than the WegenerNet JBT (Fig. 9) with its mountainous Alpine terrain.

The seasonal relative frequency of wind directions from nine-year climate data for the ZAMG Feldbach station is similar among observations and modeled values for both seasons (bottom-left panels of Fig. 9a and b).

Larger differences between modeled and observed values can be noticed for the ZAMG Bad Gleichenberg station, however

(bottom-right panels of Fig. 9a and b). For this station, the model calculates the largest fraction with about 10 to 15 % from the





NE to the E sectors for both seasons while the observed wind directions show about 17 % percent from the NNW sector and around 10 % from the S sector. These differences between modeled and observed values can be explained by the environment of this station bringing in local influences that degrade the representativeness of the wind observations for the 1-km scale (Schlager et al., 2017).

Table 7 summarizes the statistical results of multi-year seasonal mean data for selected stations including the ones illustrated in the bottom row of Fig. 9 and Fig. 10 and the ZAMG Admont station for JBT. The results of the statistical parameters generally show better performance for the WegenerNet FBR stations than for the WegenerNet JBT stations.

For the WegenerNet JBT stations the $B$ is positive for all seasons, except for the KOE station in winter. The resulting $RMSE$ ranges from about 0.9 to 1.35 m s$^{-1}$ for this station. Due to the more frequently occurring episodes of strong wind in winter,

the $RMSE$ values are generally higher for all stations in this season. Because of the higher wind speeds at the summit regions, the $RMSE$ shows higher values at a range from 2.7 to 5.1 m s$^{-1}$ for the BLA station. The $R$ value is for both the KOE and BLA stations and all seasons clearly higher than 0.6. The $MAE_{dir}$ shows for all seasons and both JBT stations similar results of near 40°.

For the ZAMG Admont station the statistical results are generally worse. Despite lower observed wind speeds compared

to the other stations, the $B$ and $RMSE$ show high values. Additionally, the $R$ and the $IOA$ values indicate poor performance, with a $R$ value only around 0.4 and $IOA$ values in a range of just -0.04 to 0.28 for all seasons. These statistical results for an independent location outside but nearby the JBT area in the Enns valley indicate the value that an additional station with wind observations also in the Enns valley could bring to the JBT network (see also Section 5 below). As noted in Sections 2 and 3, the wind observations from the existing Weidendom station, which is located in the Enns valley, are not suitable as model input

due to a non-representative location.

The WegenerNet FBR stations show a somewhat negative bias ($B$) and generally low $RMSE$ values for all seasons. The $R$ values show good results for all stations, with values higher than 0.75 throughout (ZAMG Feldbach station even > 0.85). This also applies to the $IOA$, with values higher than 0.71. The higher values of the mean absolute error of wind directions ($MAE_{dir}$) for the ZAMG Bad Gleichenberg station, compared to the ZAMG Feldbach station, indicate again the local influences affecting

the observations of this station (Schlager et al., 2017).

## 5   Conclusions

In this work we further developed an operational weather diagnostic application, the WegenerNet Wind Product Generator (WPG), recently developed by Schlager et al. (2017), and applied it to the WegenerNet Johnsbachtal (JBT), a dense meteorological station network located in a mountainous Alpine region in the north of Styria, Austria. Based on an advanced version

of the CALMET model (Scire et al., 1998), the WPG automatically generates gridded high-resolution wind fields in near-real time with a temporal resolution of 30 minutes and a spatial resolution of 100 m x 100 m. In addition, the WPG produces gridded wind gust fields with the same temporal and spatial resolution. As derived products, half-hourly fields are averaged to hourly and daily weather data products as well as monthly, seasonal and annual climate data products (Schlager et al., 2017).





A main purpose of the WPG products is the evaluation of wind fields from convection-permitting regional climate models and the investigation of weather and climate on a local scale, among other needs, such as monitoring of wind storms.

We evaluated the new WegenerNet JBT wind fields by identifying representative monthly periods with mainly thermally induced wind fields (July 2014) and strong wind speeds including wind storm events (December 2013). Using a "leave-one-
station-out" validation approach, and then evaluating against the observed wind data at the "left-out" station, we inspected the reasonableness of individual wind fields and computed statistical performance measures such as modeled vs. observed biases, root-mean-square errors and correlation coefficients. In case of wind speed, the statistics show reasonably good results for both periods with somewhat better values for December 2013. Compared to the wind speed, the analysis of wind direction delivers somewhat higher errors, with directional deviations in the wind sectors of typically around 40°, depending on the
station location and period.

Overall the results discussed well illustrate the useful level of skill, but also the evident performance limits, that the WPG application can provide for empirical wind field modeling based on a small network of seven stations in such a complex terrain as the WegenerNet JBT area.

We also evaluated seasonal statistical performance parameters for multi-year data of both the WegenerNet JBT region and
WegenerNet Feldbach region (FBR), the latter initially analyzed by Schlager et al. (2017). For the WegenerNet JBT, the statistical performance measures applied to wind speeds show reasonably good overall statistical agreement as we showed for the Koelblwiese and Blaseneck stations. The results related to wind direction show a level of directional deviation around 40°, similar to the individual month results.

For the ZAMG Admont station, an independent station nearby the area in the Enns valley, the statistics show generally poor
values, reflecting the missing meteorological wind information in the valley. The installation of an additional wind-observing station in the Enns valley (no suitable JBT station currently available there) could help to significantly improve the WPG results in this subarea. Due to the denser distribution of stations in the WegenerNet FBR, and the less challenging terrain in this Alpine foreland region, the statistical evaluation shows clearly superior climatological wind field performance for this network.

Ongoing next steps of work deal with the evaluation the dynamical wind fields of non-hydrostatic weather analyses and
climate model simulations for the two WegenerNet regions FBR and JBT for selected challenging weather contitions. For this purpose, we intercompare the empirical wind fields generated by the WPG with wind field analysis data from the INCA model of the Austrian weather service ZAMG (Haiden et al., 2011; Kann et al., 2015) as well as with climate model data from the non-hydrostatic model COSMO-CLM (Schättler et al., 2016). We expect the WPG application to be a valuable toot for serving this and other purposes.

**6   Code availability**

The CALMET 6.5.0 model code is available from the website www.src.com/calpuff/. The overall WPG code is not in the public domain and cannot be distributed.





## 7    Data availability

CORINE Land Cover data for the study area were available from www.eea.europa.eu, digital elevation model data from www.gis.steiermark.at, and WegenerNet data from www.wegenernet.org. The WegenerNet data contain the WPG wind field output data as introduced in this study on a routine basis over the entire WegenerNet data period.

5    *Author contributions.* C. Schlager collected the data, performed the analyses and modeling, created the figures, and wrote the first draft of the manuscript. Gottfried Kirchengast provided guidance and advice on all aspects of the study and significantly contributed to the text. Juergen Fuchsberger provided guidance on technical aspects of the WegenerNet networks, and its data characteristics and contributed to the text.

*Competing interests.* The authors declare that they have no conflict of interest.

10    *Acknowledgements.* The authors thank Roberto Bellasio (Enviroware), Italy, for providing the modified CALMET 5.2 code, including algorithms to account for topographic shading effects and vertical temperature gradients. WegenerNet funding is provided by the Austrian Ministry for Science and Research, the University of Graz, the state of Styria (which also included European Union regional development funds), and the city of Graz; detailed information on team, partners, and sponsors is found at www.wegcenter.at/wegenernet.





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





**Figure 1.** (a) Location of the study area WegenerNet Johnsbachtal (white-filled rectangle, enlarged in (b)) in the north of Styria, Austria. The WegenerNet Feldbach Region in the Alpine forelands of south-eastern Styria, Austria, is also indicated for reference in the easternmost part of the European Alpine region (details in Schlager et al. (2017); Figure 1 therin). (b) Map of the WegenerNet Johnsbachtal region (black rectangle) with its meteorological stations, including the selected mountain top pseudo stations, with the legend explaining map characteristics and station operators.

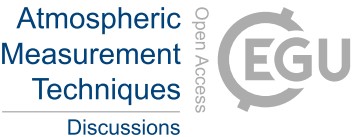

**Figure 2.** (a) Land cover and use of the WegenerNet Johnsbachtal region (black rectangle) based on the Corine Land Cover 2006 raster version. (b) Example temperature field over the the region during a summer day in July (18 July 2014; 15:00 UTC).







**Figure 3.** Modeled wind fields typical for the study area: (a) Thermally induced wind fields (18 July 2014) and (b) strong region-scale winds (07 December 2013), for near-sunrise (top), afternoon (middle) and near-sunset (bottom) conditions. Time is shown as UTC (corresponding to local time minus 1 hr).



**Figure 4.** Modeled wind gust fields typical for the study area: (a) Thermally induced wind fields (18 July 2014) and (b) strong region-scale winds (07 December 2013), for near-sunrise (top), afternoon (middle) and near-sunset (bottom) conditions. Time is shown as UTC (corresponding to local time minus 1 hr).


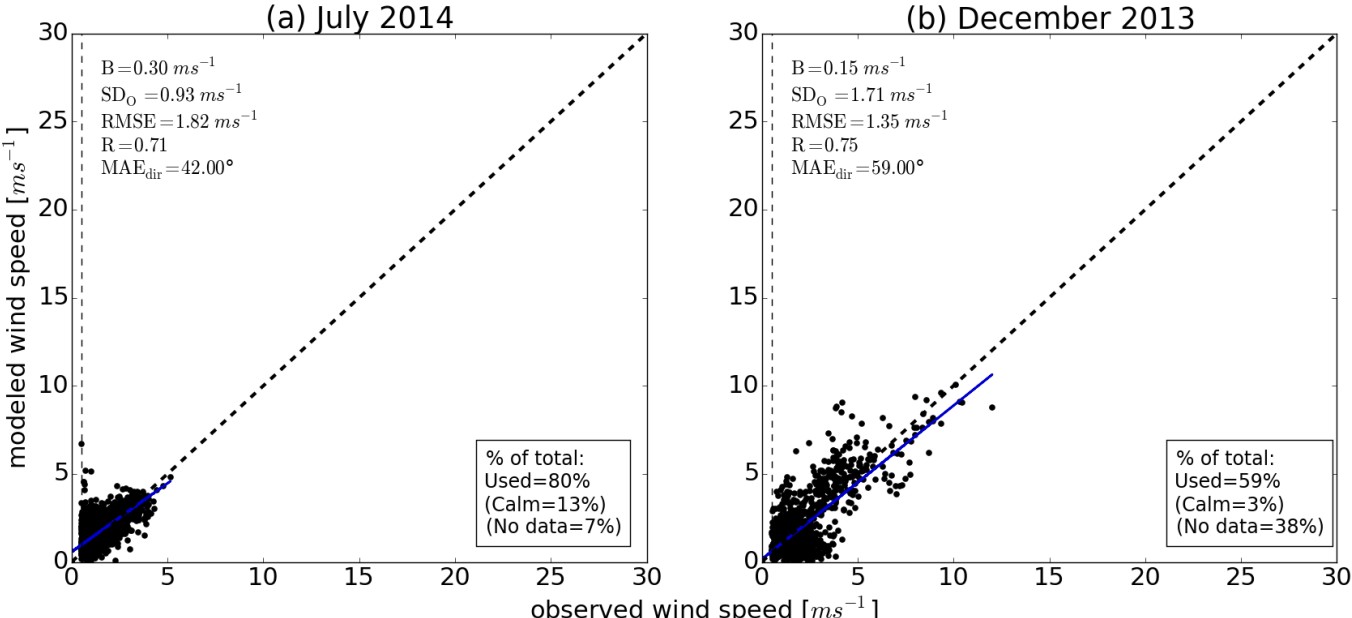

**Figure 5.** Scatterplot of modeled versus observed vector-mean wind speeds for the WegnerNet Koelblwiese (KOE) station in the Johnsbach valley: (a) July 2014 and (b) December 2013.

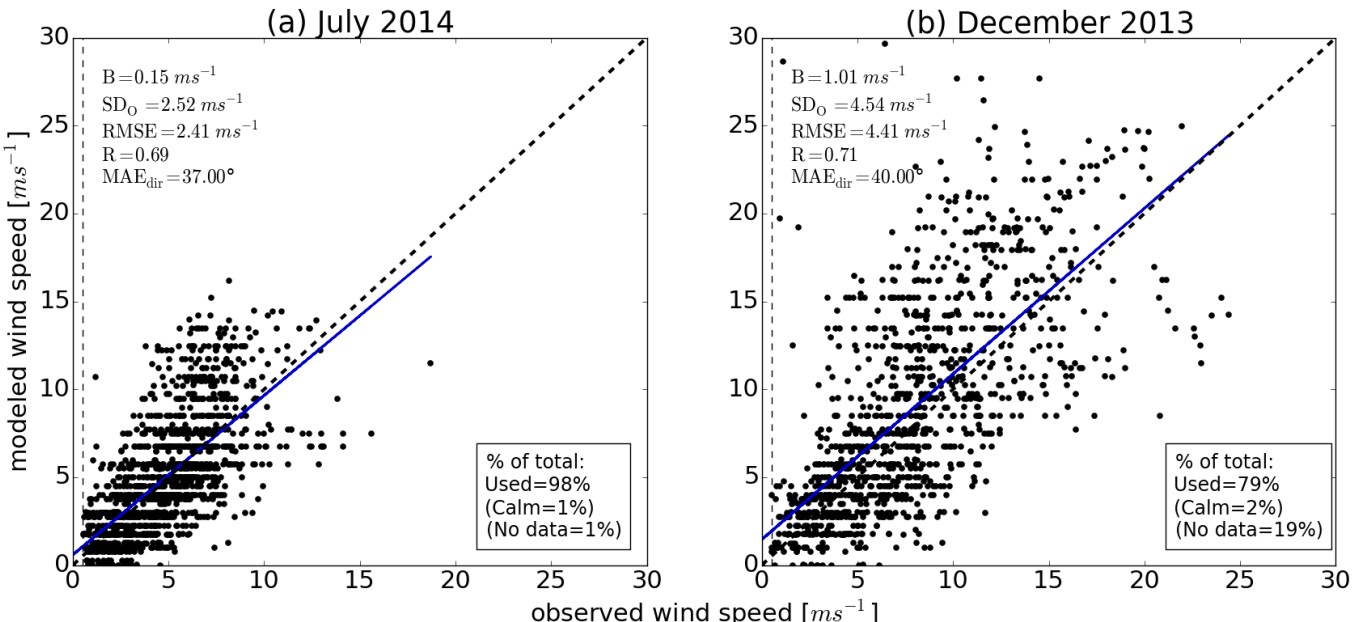

**Figure 6.** Same as Fig. 5 but for WegenerNet Blaseneck (BLA) station at the Blaseneck summit.





**Figure 7.** Relative frequency of vector-mean wind directions for a range of wind speed categories, for observed (upper row) and modeled (lower row) wind directions for the WegenerNet Koelblwiese (KOE) station in the Johnsbach valley: (a) July 2014 and (b) December 2013.



**Figure 8.** Same as Fig. 7 but for WegenerNet Blaseneck (BLA) station at the Blaseneck summit.





**Figure 9.** Modeled five-year/four-year seasonal mean wind fields (maps, top) and relative frequency of wind directions for the Koelblwiese (KOE) and Blaseneck (BLA) station (windroses, bottom) for the WegenerNet JBT: (a) Summer month 03/2012/(03/2013)-02/2017 and (b) Winter month 03/2012/(03/2013)-02/2017.





**Figure 10.** Same as Fig. 9 but for nine-year seasonal means in the WegenerNet FBR, and windrose results for the ZAMG Feldbach and Bad Gleichenberg stations.




**Table 1.** Characteristics of meteorological stations of the WegenerNet Johnsbachtal (JBT).

| Station name, ID (start[a]) | Station abbreviation | Operator | Lat (E) | Lon (N) | Alt [m] | Wind sensor height [m] | Recorded variables[b] |
|---|---|---|---|---|---|---|---|
| Oberkainz, 501 (2010) | OBK | WEGC | 47° 32' 11.0" | 14° 35' 52.8" | 920 | 14 | $v$, $\phi$, $v_g$, $\phi_g$, $T$, $rh$, $P$, $Q_g$, $Q_r$, $sd$, $swe$ |
| Koelblwiese, 502 (2013) | KOE | WEGC | 47° 31' 54.0" | 14° 36' 37.0" | 860 | 3 | $v$, $\phi$, $v_g$, $\phi_g$, $T$, $rh$, $P$, $p$, $Q_g$, $Q_r$, $Q_n$ |
| Schroeckalm, 503 (2010) | SCH | WEGC | 47° 31' 45.2" | 14° 40' 16.8" | 1344 | 10 | $v$, $\phi$, $v_g$, $\phi_g$, $T$, $rh$, $P$, $p$, $Q_g$, $Q_n$, $\rho_s$ |
| Blaseneck, 504 (2010) | BLA | WEGC | 47° 29' 57.7" | 14° 37' 07.9" | 1969 | 6 | $v$, $\phi$, $v_g$, $\phi_g$, $T$, $rh$, $Q_g$, $Q_n$ |
| Zinoedl, 505 (2009) | ZIN | WEGC | 47° 33' 55.4" | 14° 39' 57.8" | 2191 | 6 | $v$, $\phi$, $v_g$, $\phi_g$, $T$, $rh$, $Q_g$, $Q_n$ |
| Weidendom, 506 (2006) | WEI | NPG | 47° 34' 51.0" | 14° 35' 29.3" | 590 | 2 | $v$, $T$, $h$, $P$, $Q_g$, $wl$ |
| Gscheidegg, 507 (2008) | GSC | NPG | 47° 30' 52.0" | 14° 40' 28.2" | 1690 | 6 | $v$, $\phi$, $v_g$, $\phi_g$, $T$, $rh$, $p$, $Q_g$, $sd$, $\rho_s$ |
| Tamischb. 1, 508 (2008) | TA1 | ZAWS | 47° 37' 02.4" | 14° 43' 01.2" | 1431 | 7 | $v$, $\phi$, $v_g$, $\phi_g$, $T$, $rh$, $Q_g$, $T_s$, $sd$, $T_{sn}$ |
| Tamischb. 2, 509 (2008) | TA2 | ZAWS | 47° 36' 48.4" | 14° 41' 58.2" | 1952 | 5 | $v$, $\phi$, $v_g$, $\phi_g$, $T$, $rh$ |
| Gstatterboden, 510 (2007) | GST | AHYD | 47° 35' 29.0" | 14° 37' 44.0" | 580 | - | $P$ |
| Gaishorn, 511 (2007) | GAI | AHYD | 47° 35' 29.0" | 14° 37' 44.0" | 720 | - | $P$ |

[a] start year of time series (earliest year in WegenerNet archive is 2007)

[a] $v$ wind speed, $\phi$ wind direction, $v_g$ peak gust, $\phi_g$ peak gust direction, $T$ air temperature, $rh$ relative humidity, $P$ precipitation, $p$ air pressure , $sd$ snow depth, $swe$ snow water equivalent, $\rho_s$ snow density, $Q_g$ global radiation, $Q_r$ reflected radiation, $Q_n$ net radiation, $T_s$ surface temperature, $T_{sn}$ snow temperature, and $wl$ water level



**Table 2.** Settings of critical area-specific model parameters in CALMET, used in this study for the WegenerNet JBT.

| Parameter | Value | Remarks |
|---|---|---|
| TERRAD [km] | 5.0 | Radius of influence of terrain features |
| RMAX1 [km] | 5.0 | Maximum radius of influence over land in the surface layer |
| RMAX2 [km] | 5.0 | Maximum radius of influence over land aloft |
| R1 [km] | 1.1 | Relative weighting of the first guess field and observations in the surface layer |
| R2 [km] | 0.6 | Relative weighting of the first guess field and observations in the layers aloft |
| IEXTRP (flag) | -4 | Extrapolate surface wind observations to upper layers with similarity theory (layer 1 data at upper air stations are ignored) |
| ZFACE [m] | 0, 20, 80 | Cell face heights in vertical grid (the vertical levels correspond to the mid-levels, 10 m and 50 m, of those layer boundaries) |
| BIAS ($-1 \leq$ BIAS $\leq 1$) | 0.0, 0.0, 0.0, 0.0 | Layer-dependent biases modifying the weights of surface and upper air stations (Negative BIAS reduces the weight of upper air stations, positive BIAS reduces the weight of surface stations) |



**Table 3.** Geophysical parameters based on the Corine Land Cover (CLC) 2006 dataset, used in this study for the WegenerNet JBT.

| *Land use type* | *Surface roughness length* [m] | *Albedo* | *Bowen ratio* | *Soil heat flux constant* | *Vegetative leaf area index* |
|---|---|---|---|---|---|
| Discontinuous urban fabric | 1.000 | 0.18 | 1.5 | 0.25 | 0.20 |
| Agricultural land - unirrigated | 0.250 | 0.15 | 1.0 | 0.15 | 3.00 |
| Rangeland | 0.050 | 0.25 | 1.0 | 0.15 | 0.50 |
| Forest land | 1.000 | 0.10 | 1.0 | 0.15 | 7.00 |
| Small water body | 0.001 | 0.10 | 0.0 | 1.00 | 0.00 |
| Nonforest wetland | 0.020 | 0.10 | 0.10 | 0.25 | 1.00 |
| Barren land | 0.050 | 0.30 | 1.0 | 0.15 | 0.05 |

**Table 4.** Characteristics of pseudo upper-air stations defined in the WegenerNet JBT region.

| *Station name* | *Station abbreviation* | *Latitude* (E) | *Longitude* (N) | *Altitude* [m] | *Neighbor stations* (*Table* 1) |
|---|---|---|---|---|---|
| Pseudo station 1 | PS1 | 47° 36' 49.5" | 14° 36' 06.0" | 2061 | TA2; ZIN |
| Pseudo station 2 | PS2 | 47° 32' 59.6" | 14° 31' 24.9" | 2126 | BLA; ZIN |
| Pseudo station 3 | PS3 | 47° 33' 36.9 | 14° 37' 45.0"" | 2068 | BLA; ZIN |
| Pseudo station 4 | PS4 | 47° 33' 16.0" | 14° 43' 33.7" | 2139 | BLA; ZIN |
| Pseudo station 5 | PS5 | 47° 29' 02.1" | 14° 42' 06.3" | 1892 | BLA; ZIN |





**Table 5.** Statistical performance parameters used for the evaluation of the wind field modeling results.

| Parameter | Equation | Remarks |
|---|---|---|
| Bias | $B = \frac{1}{N} \sum_{i=1}^{N} (v_{m,i} - v_{o,i})$ | $v_m$: modeled wind speed; $v_o$: observed wind speed |
| Standard dev. of observed wind speed | $SD_o = \sqrt{\frac{1}{(N-1)} \sum_{i=1}^{N} (v_{o,i} - \overline{v_o})^2}$ | $v_o$: observed wind speed; $\overline{v_o}$: mean observed wind speed |
| Root-mean-square error | $RMSE = \sqrt{\frac{1}{N} \sum_{i=1}^{N} (v_{m,i} - v_{o,i})^2}$ | $v_m$: modeled wind speed; $v_o$: observed wind speed |
| Correlation coefficient | $R = \frac{1}{(N-1)} \sum_{i=1}^{N} \left( \frac{v_{m,i} - \overline{v_m}}{\sigma_m} \right) \left( \frac{v_{o,i} - \overline{v_o}}{\sigma_o} \right)$ | $v_m$: modeled wind speed; $\overline{v_m}$: mean modeled wind speed; $v_o$: observed wind speed; $\overline{v_o}$: mean observed wind speed; $\sigma_m$: standard deviation of modeled wind speed; $\sigma_o$: standard deviation of observed wind speed |
| Index of agreement | $IOA = \begin{cases} 1.0 - \frac{\sum_{i=1}^{N} \left| v_{m,i} - v_{o,i} \right|}{c \sum_{i=1}^{N} \left| v_{o,i} - \overline{v_o} \right|}, & \text{if } \sum_{i=1}^{N} \left| v_{m,i} - v_{o,i} \right| \leq c \left| v_{o,i} - \overline{v_o} \right| \\[2ex] \frac{c \sum_{i=1}^{N} \left| v_{o,i} - \overline{v_o} \right|}{\sum_{i=1}^{N} \left| v_{m,i} - v_{o,i} \right|} - 1, & \text{if } \sum_{i=1}^{N} \left| v_{m,i} - v_{o,i} \right| > c \left| v_{o,i} - \overline{v_o} \right| \end{cases}$ | $v_m$: modeled wind speed; $v_o$: observed wind speed; $\overline{v_o}$: mean observed wind speed; $c$: factor set to 2 (Willmott et al., 2012) |
| Mean absolute error of wind direction | $MAE_{dir} = \frac{1}{N} \sum_{i=1}^{N} \{ \arccos[\cos(\phi_{m,i} - \phi_{o,i})] \}$ | $\phi_m$: modeled wind direction; $\phi_o$: observed wind direction |



**Table 6.** Statistical performance measures calculated for the WegenerNet JBT meteorological stations with contributing wind sensors, for July 2014 and December 2013 from the "leave-one-out" validation analysis. See Table 5 for more information on the calculation of the performance parameters.

| Station ID and Abbr. | July 2014 | | | | | | December 2013 | | | | | |
|---|---|---|---|---|---|---|---|---|---|---|---|---|
| | $B$ [m s$^{-1}$] | $SD_o$ [m s$^{-1}$] | $RMSE$ [m s$^{-1}$] | $R$ [1/1] | $IOA$ [1/1] | $MAE_{dir}$ [°] | $B$ [m s$^{-1}$] | $SD_o$ [m s$^{-1}$] | $RMSE$ [m s$^{-1}$] | $R$ [1/1] | $IOA$ [1/1] | $MAE_{dir}$ [°] |
| 501, OBK | -0.10 | 1.23 | 1.24 | 0.42 | 0.57 | 35 | 1.61 | 1.71 | 3.72 | 0.35 | 0.21 | 51 |
| 502, KOE | 0.30 | 0.93 | 0.81 | 0.71 | 0.61 | 42 | -0.15 | 1.71 | 1.35 | 0.75 | 0.56 | 59 |
| 503, SCH | 0.67 | 0.89 | 1.82 | 0.39 | 0.25 | 54 | 1.45 | 1.59 | 3.32 | 0.61 | 0.22 | 40 |
| 504, BLA | 0.15 | 2.52 | 2.41 | 0.69 | 0.55 | 37 | 1.01 | 4.54 | 4.41 | 0.71 | 0.55 | 40 |
| 505, ZIN | -0.67 | 3.44 | 2.56 | 0.70 | 0.66 | 36 | -3.85 | 6.76 | 6.02 | 0.73 | 0.60 | 38 |
| 507, GSC | 0.31 | 1.01 | 1.10 | 0.56 | 0.46 | 74 | 0.83 | 1.28 | 1.85 | 0.62 | 0.32 | 67 |
| 509, TA2 | 1.40 | 2.47 | 3.01 | 0.62 | 0.46 | 50 | 0.99 | 4.52 | 4.62 | 0.69 | 0.54 | 37 |
| *Mean Value* | 0.30 | 1.78 | 1.85 | 0.58 | 0.51 | 47 | 0.27 | 3.16 | 3.61 | 0.64 | 0.43 | 47 |





**Table 7.** Statistical multi-year climatological performance measures calculated for representative meteorological stations for the WegenerNet JBT and the WegenerNet FBR (upper half five-year/four-year seasonal means for three WegenerNet JBT stations; right half nine-year seasonal means for two WegenerNet FBR stations). See Table 5 for more information on the calculations of the performance parameters.

| Season (per Station) | $B$ [m s$^{-1}$] | $SD_o$ [m s$^{-1}$] | $RMSE$ [m s$^{-1}$] | $R$ [1/1] | $IOA$ [1/1] | $MAE_{dir}$ [°] |
|---|---|---|---|---|---|---|
| *WegenerNet JBT* | | | | | | |
| KOE: 03/2013-02/2017 | | | | | | |
| spring (MAM) | 0.18 | 1.50 | 1.06 | 0.75 | 0.68 | 39 |
| summer (JJA) | 0.25 | 1.16 | 0.89 | 0.75 | 0.67 | 38 |
| fall (SON) | 0.16 | 1.35 | 1.10 | 0.68 | 0.63 | 41 |
| winter (DJF) | -0.17 | 1.57 | 1.35 | 0.67 | 0.58 | 47 |
| all | 0.13 | 1.41 | 1.09 | 0.71 | 0.65 | 41 |
| BLA: 03/2012-02/2017 | | | | | | |
| spring (MAM) | 0.09 | 3.54 | 3.64 | 0.65 | 0.51 | 40 |
| summer (JJA) | 0.34 | 2.70 | 2.74 | 0.68 | 0.54 | 43 |
| fall (SON) | 0.74 | 3.50 | 3.67 | 0.67 | 0.52 | 39 |
| winter (DJF) | 0.04 | 4.91 | 5.09 | 0.64 | 0.54 | 41 |
| all | 0.73 | 3.63 | 3.65 | 0.67 | 0.54 | 41 |
| ZAMG ADM[a]: 03/2012-02/2017 | | | | | | |
| spring (MAM) | 1.33 | 1.38 | 3.28 | 0.38 | 0.19 | 52 |
| summer (JJA) | 0.99 | 1.18 | 2.62 | 0.36 | 0.28 | 53 |
| fall (SON) | 1.17 | 1.15 | 2.89 | 0.47 | 0.07 | 40 |
| winter (DJF) | 1.38 | 1.09 | 3.59 | 0.43 | -0.04 | 36 |
| all | 1.22 | 1.22 | 3.12 | 0.40 | 0.15 | 38 |
| *WegenerNet FBR* | | | | | | |
| ZAMG FB[b]: 03/2008-02/2017 | | | | | | |
| spring (MAM) | -0.28 | 1.36 | 0.75 | 0.86 | 0.78 | 22 |
| summer (JJA) | -0.27 | 1.00 | 0.57 | 0.87 | 0.77 | 19 |
| fall (SON) | -0.25 | 1.05 | 0.57 | 0.87 | 0.78 | 19 |
| winter (DJF) | -0.21 | 1.07 | 0.54 | 0.89 | 0.80 | 16 |
| all | -0.25 | 1.15 | 0.61 | 0.88 | 0.79 | 19 |
| ZAMG BG[c]: 03/2008-02/2017 | | | | | | |
| spring (MAM) | -0.17 | 1.22 | 0.83 | 0.76 | 0.71 | 31 |
| summer (JJA) | -0.08 | 0.92 | 0.64 | 0.76 | 0.71 | 57 |
| fall (SON) | -0.12 | 0.88 | 0.60 | 0.77 | 0.73 | 27 |
| winter (DJF) | -0.11 | 0.87 | 0.57 | 0.79 | 0.73 | 26 |
| all | -0.12 | -1.00 | 0.67 | 0.78 | 0.72 | 28 |

[a]Admont station, [b]Feldbach station, [c]Bad Gleichenberg station