# Peer review of "Empirical high-resolution wind field and gust model in mountainous and hilly terrain based on the dense WegenerNet station networks"

_Atmospheric Measurement Techniques, 2018_

## Referee Comment (RC1) · Anonymous Referee #1 · 26 Jul 2018

GENERAL COMMENTS The authors describe the results of a near-real time modeling tool developed for the generation of gridded wind field over complex terrain in Austria. The tool is based on the measurements of a network of meteorological stations and on a modified version of the CALMET model. The work is based on a valid scientific approach, the paper is well structured and the results clearly described.

SPECIFIC COMMENTS Page 1, lines 11-13. The authors should note that strong winds tend to have an almost constant direction, while weak winds are often characterized by variable directions. Therefore strong winds are relatively more easy to predict. Page 6, Lines 10-11: The sentence is not clear. Please

reformulate. Page 6, equation 3: Note that there are more compact equations to get the wind direction starting from the wind components. See for example: https://www.researchgate.net/profile/Stuart_Grange2/publication/262766424_Technical_note note-Averaging-wind-speeds-and-directions.pdf Page 8, line 13: where $\varphi$g is the direction of the peak gust speed and $\varphi$m the 30-min vector-mean wind direction. Figures 5 and 6: Text within figures is very small.

TECHNICAL CORRECTIONS Page 1 (Abstract), line 6: 100 x 100 m2 Page 1 (Abstract), line 7: The main purpose (not "A main purpose..."). Page 2, line 20: ... characterized by a very complex terrain. Page 3, line 33: Sometimes a dot is used to separate thousands, other times not. Please use the same rule in the whole paper. Page 5, line 28: Fig. 1b Page 9, line 20: ... occurred at the same time ... Page 13, line 13: The main purpose (not "A main purpose..."). Page 14, line 7: ... valuable tool ...

---

## Author Comment (AC1) · 10 Aug 2018

We would like to thank you again for your feedback to our manuscript and will implement your proposed changes.

Answerers to your specific comments: 1.) Page 1, lines 11-13. The authors should note that strong winds tend to have an almost constant direction, while weak winds are often characterized by variable directions. Therefore strong winds are relatively more easy to predict. Answer: Ok, we will improve the description to clarify that strong wind speeds are easier to model in the mountainous region of the Johnsbachtal. Also it has to be noted that the modeling performance shows opposite values for the hilly

Feldbach region, with somewhat better values for weak wind speed events than for strong wind speed events (Schlager et. al. 2017). Improved description: "The overall statistical agreement, estimated for the vector-mean wind speed, shows a reasonably good modeling performance. Due to the spatially more homogeneous wind speeds and directions for strong wind events in this mountainous region, the results show somewhat better performance for these events."

2.) Page 6, Lines 10-11: The sentence is not clear. Please reformulate. Answer: Ok, we will improve the description related to the estimation of the magnitude of wind speed of a pseudo station. Improved description: "The magnitude of wind speed of a pseudo station (vp) is estimated for the highest defined vertical height level (zmax), which corresponds to the highest ZFACE level (Table 2; 80 m). The estimation is based on a linear interpolation between neighbor station altitudes, except for pseudo station 5, which is located at somewhat lower altitude than its neighborhood stations. The wind speed is hence calculated by a slight downward extrapolation for this latter station."

3.) Page 6, equation 3: Note that there are more compact equations to get the wind direction starting from the wind components. See for example: https://www.researchgate.net/profile/Stuart_Grange2/publication/262766424_Technical_note_Averaging_wind_speeds_and note-Averaging-wind-speeds-and-directions.pdf Answer: Thank you for this hint, we agree this is more compact and looks more elegant. We will therefore change to this formulation from your suggested paper.

4.) Figures 5 and 6: Text within figures is very small. Answer: Thank you, ok, we will further increase the font size of the text of Figure 5 and 6, especially in the in-panel legends at upper left in the panels of these figures (which we agree are a bit small).

Answer regarding technical corrections: 5.) Thank you for the care related to these remaining typos or spelling mistakes. We will account for these technical-editorial improvement suggestions that you listed.

[Figure]

Please also note the supplement to this comment:
https://www.atmos-meas-tech-discuss.net/amt-2018-31/amt-2018-31-AC1-
supplement.pdf

---

## Referee Comment (RC2) · Anonymous Referee #1 · 16 Aug 2018

Dear Authors, thank you for your answers. The new version of the manuscript is fine with me. I have no additional comments.

---

## Referee Comment (RC3) · Anonymous Referee #3 · 5 Sep 2018

Dear Authors, please take into consideration the suggestion from Referee #2:

Suggestions for technical corrections: *In the statistical analysis of the paper, I suggest changing the correlation coefficient R by the coefficient of determination $R^2$.

*On page 10, between lines 20-25, which means the concept of "good accordance", how this concept evaluates from the statistical point of view. I suggest calculating, because it is very simple to do so, to use some "agreement index", such as Index of Agreement (d) developed by Willmott (1981).

Willmott, C. J. 1981. On the validation of models. Physical Geography, 2, 184-194

[Figure]

With this index I believe that the concept of "good accordance" can be applied and discussed in the article.

*When discussing the results in terms of the RMSE, please indicate some kind of qualitative qualification, for example: high, medium or low.

*Please, in the article, mention how the effect of the pixel size of the DEM can affect the results of the modeling. In addition, the modification made by the authors to the CALMET model to improve the estimation of solar radiation, carried out previously in another article, also affects the data of the modeling, however it is not well developed in the article. I would have expected a comparison between the results of the unmodified CALMET model and the modified CALMET model, using some agreement index like the Akaike information criterion (AIC).

Between lines 11-12, this paragraph should be rewritten.

The sources of error are not adequately evaluated in the conclusions. Again, "good agreement" is mentioned, without having calculated any index of the literature that accounts for this concept.

---

## Editor Comment (EC1) · L. Bianco (Editor) · 6 Sep 2018

Dear Authors, please find below the final comment from Referee#2:

Dear authors, thanks for answering about my suggestions. I think the current version is fine, I have no additional suggestions.

[Figure]

---

## Author Comment (AC2) · 6 Sep 2018

Thank you again for your feedback to our manuscript

---

## Author Response (AR1)

**Response to Reviewer #1 and Reviewer #2**

of " Empirical high-resolution wind field and gust model in mountainous and hilly terrain based on the dense WegenerNet station networks"
by C Schlager, G Kirchengast and J Fuchsberger. Submitted to AMT, January 2018;

*We thank the Reviewers again very much for the valuable and quite detailed feedback to our manuscript. We carefully considered all comments and made due effort to account for the concerns expressed; and we think it really helped improving the comprehensibility and quality of the text and how we convey the findings. We also would like to thank the Reviewers for the care also related to remaining typos and spelling mistakes. We corrected in line with all of these suggestions.*
*Comments by the Reviewer are* black upright*, our responses blue italic. (Page and line numbers used in our responses below refer to the revised manuscript; to make this clear they are quoted like "now p10 L20-25")*

**Response to Reviewer # 1 from first submission**

1 In some figures, for example from fig 5 to fig 9, the internal text is very small and may be difficult to read. In some of these figures also, font changes within the same sentence.
*Answer: Thank you for this hint; we agree that the font size of the internal text in some figures was somewhat too small. We therefore increased the font size of the internal text from Figure 5 to Figure 10. We also increased the font size of the axes labels of the wind roses from Figure 7 to Figure 10.*

**Response to Reviewer # 1 from interactive discussion**

Answerers to your specific comments:
1 Page 1, lines 11-13. The authors should note that strong winds tend to have an almost constant direction, while weak winds are often characterized by variable directions. Therefore strong winds are relatively more easy to predict.
*Answer: Ok, we improved the description to clarify that strong wind speeds are easier to model in the mountainous region of the Johnsbachtal (now p1 L12-14).*
*Also it has to be noted that the modeling performance shows opposite values for the hilly Feldbach region, with somewhat better values for weak wind speed events than for strong wind speed events (Schlager et. al. 2017). Improved description: "The overall statistical agreement, estimated for the vector-mean wind speed, shows a reasonably good modeling performance. Due to the spatially more homogeneous wind speeds and directions for strong wind events in this mountainous region, the results show somewhat better performance for these events."*

2 Page 6, Lines 10-11: The sentence is not clear. Please reformulate.
*Answer: Ok, we improved the description related to the estimation of the magnitude of wind speed of a pseudo station (now p6 L14-16).*

3 Page 6, equation 3: Note that there are more compact equations to get the wind direction starting from the wind components. See for example: https://www.researchgate.net/profile/Stuart_Grange2/publication/262766424_Technical_note_Averaging_wind_speeds_and_directions/links/54f6184f0cf27d8ed71d5bd4/Technical-note-Averaging-wind-speeds-and-directions.pdf.

*Answer: Thank you for this hint, we agree this is more compact and looks more elegant. We therefore changed to this formulation from your suggested paper (now p6 L26). Based on these equations, the wind direction is calculated from the south and west, instead of north and east components. We therefore adapted the equations for the calculation of these components (now p7 L3 and L5) and the corresponding text (now p7 L1-2).*

4 Figures 5 and 6: Text within figures is very small.

*Answer: Thank you, ok, we further increased the font size of the text of Figure 5 and 6, especially in the in-panel legends at upper left in the panels of these figures (which we agree are a bit small). Answer regarding technical corrections: Thank you for the care related to these remaining typos or spelling mistakes. We accounted for these technical-editorial improvement suggestions that you listed (details below).*

Answer regarding technical corrections:
5 Page 1 (Abstract), line 6: 100 x 100 m2 Page 1 (Abstract)
*Answer: OK, done (now p1 L6)*

6 line 7: The main purpose (not "A main purpose")
*Answer: OK, done (now p1 L8)*

8 Page 2, line 20: characterized by a very complex terrain. Page 3
*Answer: OK, done (now p2 L20)*

9 Page 3, line 33: Sometimes a dot is used to separate thousands, other times not. Please use the same rule in the whole paper
*Answer: OK, we removed the dots (now p3 L33, p4 L1, L4, L12)*

10 Page 5, line 28: Fig. 1b
*Answer: OK, done (now p5 L28)*

11 Page 9, line 20: occurred at the same time
*Answer: OK, done (now p9 L29)*

12 Page 13, line 13: The main purpose (not "A main purpose").
*Answer: OK, done (now p13 L27)*

13 Page 14, line 7: valuable tool
*Answer: OK, done (now p14 L21)*

**Response to Reviewer # 2 from first submission**

1 In the statistical analysis of the paper, I suggest changing the correlation coefficient R by the coefficient of determination r2.
*Answer: Thank you for this hint; we understand that the R-squared is sometimes preferred in these evaluation methods and carefully considered to change this. To avoid inconsistency with our Schlager et al. 2017 paper published to WAF, however, we preferred to keep it in the case of this paper as the correlation coefficient R.*

2 On page 10, between lines 20-25, which means the concept of "good accordance", how this concept evaluates from the statistical point of view. I suggest calculating, because it is very simple to do so, to use some "agreement index", such as Index of Agreement (d) developed by Willmott (1981).
Willmott, C. J. 1981. On the validation of models. Physical Geography, 2, 184-194
With this index I believe that the concept of "good accordance" can be applied and discussed in the article.
*Answer: In this context we used the term "good accordance" just for the visual interpretation and explanation of Figure 8. To clarify the statements, we improved the wording of the sentences for the description of this Figure (now p10 L32-33 and p11 L4-5).*
*Thank you for your suggestion regarding the Index of Agreement (IOA). We agree, and in fact we calculated it in our Schlager et al. 2017 paper as well. To be consistent with this paper, we use the redefined IOA of Wilmott (2002) and added a description about this parameter to the manuscript (now p9 L4-9 and table 5). We now also discuss the calculated IOA values in the results section (now p11 L16-18, p13 L8-9, p13 L15-16).*

3 When discussing the results in terms of the RMSE, please indicate some kind of qualitative qualification, for example: high, medium or low.
*Answer: Thank you for this proposal; we implemented some text changes to give some qualitative qualification (now p11 L11-12, p13 L2-3, p13 L14).*

4 Please, in the article, mention how the effect of the pixel size of the DEM can affect the results of the modeling. In addition, the modification made by the authors to the CALMET model to improve the estimation of solar radiation, carried out previously in another article, also affects the data of the modeling, however it is not well developed in the article. I would have expected a comparison between the results of the unmodified CALMET model and the modified CALMET model, using some agreement index like the Akaike information criterion (AIC).
*Answer: OK, we agree the description related to the DEM was a bit crude. Therefore we added a paragraph to the manuscript, which explains the performed sensitivity tests regarding different spatial resolutions (now p5 L28-33).*
*We emphasize that the main motivation why we modified the original CALMET was the generation of overly simplified temperature fields by this original model. The original CALMET produces these fields by a simple (horizontal) interpolation of point-specific temperature observations. Especially in the JBT, with its large differences in altitude, large temperature gradients can occur, however, which may affect the wind field and should hence be allowed for. Because auf this and since it creates no other disadvantages, we used the modified CALMET, since it produces a more realistic temperature field accounting for vertical gradients, which are estimated from the*

*range of meteorological stations located at different altitudes. The algorithms further empirically take into account the shading through the relief based on the DEM and the leaf area index. An example of such a generated temperature field is illustrated in Figure 2.*
*We reconsidered also our description related to this; we think that the description of our motivation for using the modified version is already detailed enough (see p5 L11-L21).*

5 Between lines 11-12, this paragraph should be rewritten.
*Answer: OK, could you please indicate the page number. We checked through the pages but were not sure which page and hence which text-piece was perhaps meant. We will improve this paragraph in our final version of the manuscript as needed.*

6 The sources of error are not adequately evaluated in the conclusions. Again, "good agreement" is mentioned, without having calculated any index of the literature that accounts for this concept.
*Answer: Thank you for this hint; with this statement we refer to all statistical performance measures applied to wind speeds (B, $SD_o$, RMSE, R, and now also IOA). To avoid ambiguities we replaced "statistical agreement" with "overall statistical agreement" in the manuscript, in order to better express it is a type of summarizing statement (now p1 L11-12 and p14 L8-10). We similarly did that in Schlager et al. 2017.*

**Further changes in the manuscript**

*1 Changed word order in a sentence on p1 (now p1 L20-21)*

*2 We added additional text regarding a further improvement of the generation of gridded fields of peak gust speed to avoid unrealistic high gust speeds, especially under calm weather conditions (now p8 L7-9). The results shown in Figure 4 are not influenced by this incremental improvement to the gust modeling in general.*

[revised manuscript text omitted]